# GLIS1 in Cancer-Associated Fibroblasts Regulates the Migration and Invasion of Ovarian Cancer Cells

**DOI:** 10.3390/ijms23042218

**Published:** 2022-02-17

**Authors:** Mi Joung Kim, Daun Jung, Joo Youn Park, Seung Min Lee, Hee Jung An

**Affiliations:** 1Institute for Clinical Research, CHA Bundang Medical Center, CHA University, Seongnam 13496, Korea; mjkim7845@naver.com (M.J.K.); jypark@naver.com (J.Y.P.); smin0515@chamc.co.kr (S.M.L.); 2Department of Pathology, CHA Bundang Medical Center, CHA University, Seongnam 13496, Korea; jhd2800@hanmail.net

**Keywords:** ovarian cancer, cancer-associated fibroblast, reprogramming, metastasis, GLIS1

## Abstract

A cancer-associated fibroblasts (CAFs) are the most important players that modulate tumor aggressiveness. In this study, we aimed to identify CAF-related genes in ovarian serous carcinomas (OSC) that account for the high incidence and mortality of ovarian cancers (OCs) and to develop therapeutic targets for tumor microenvironment modulation. Here, we performed a microarray analysis of CAFs isolated from three metastatic and three nonmetastatic OSC tissues and compared their gene expression profiles. Among the genes increased in metastatic CAFs (mCAFs), GLIS1 (Glis Family Zinc Finger 1) showed a significant increase in both the gene mRNA and protein expression levels. Knockdown of GLIS1 in mCAFs significantly inhibited migration, invasion, and wound healing ability of OC cells. In addition, an in vivo study demonstrated that knockdown of GLIS1 in CAFs reduced peritoneal metastasis. Taken together, these results suggest that CAFs support migration and metastasis of OC cells by GLIS1 overexpression. It also indicates GLIS1 in CAFs might be a potential therapeutic target to inhibit OC metastasis.

## 1. Introduction

The tumor microenvironment (TME) is a milieu in each tumor composed of different cellular and structural factors, including blood vessels, immune cells, stromal cells, and the extracellular matrix, and is involved in either tumor promotion or regression. Cancer-associated fibroblasts (CAFs), among the stromal components, are the most important players involved in modulating the TME and influencing aggressive tumor progression [1,2,3]. CAFs are reportedly associated with tumor metastases and invasion [1,2], and evidence has demonstrated that CAFs represent the major players in tumor–stroma crosstalk in the TME and enhance tumor progression by promoting angiogenesis or lymphangiogenesis. The role of CAFs in tumor progression, the regulation of the response to therapies, and the prognostic relevance of markers associated with CAFs in different tumors have been recently studied [4,5]. In addition, CAFs have phenotypic and functional heterogeneity [4,5,6], indicating the existence of different subpopulations with distinct functions in the TME.

Based on these findings, many investigators have sought to develop therapeutics targeting CAFs in the TME. Several reports have shown that the downregulation of CAF-mediated genes decreases the metastasis and growth of various human tumors, including lung cancer, breast cancer, and ovarian cancer (OC) [7,8,9,10].

OC is the most lethal gynecologic malignancy, and its mortality rate has not improved due to frequent recurrence after primary therapy and the lack of treatment after disease relapse [3]. Thus, it is important to understand the role of the surrounding tumor microenvironment (TME), especially in CAFs, which regulate tumor progression, to develop better therapeutic strategies for overcoming the tumorigenic effect of the TME in OC.

In this study, to identify metastatic CAF (mCAF)-related genes in OC and to identify new therapeutic targets for this tumor, we examined the differences in gene expression profiles between (mCAFs) and nonmetastatic CAFs (nmCAFs) in the TME of ovarian serous carcinoma (OSC), which accounts for the highest incidence and the most lethal subtype among all OCs. We further investigated the functions of the significant differential expressed genes (DEGs) between mCAFs and nmCAFs.

## 2. Results

### 2.1. Gene Expression Profile of Primary mCAFs and nmCAFs Isolated from the Cancer Tissues of OSC Patients

To investigate the gene profiles in normal fibroblasts (NOFs), mCAFs, and nmCAFs from primary OSC tissues, we performed a microarray analysis. To define the genes involved in tumor metastases and invasion, we transcriptionally compared sorted mCAFs and nmCAFs. Positive signals were obtained from 22,155 clones hybridized with probes. A total of 129 genes were differentially expressed, of which 69 genes were upregulated and 60 genes were downregulated on mCAFs as compared to nmCAFs (fold change (FC) > 2 or <0.5, *p* < 0.05) (Figure 1A,B).

To exclude the genes upregulated in NOFs, we compared mCAFs and NOFs. Sixty-two genes were significantly higher in mCAFs than in NOFs. Among these genes, five genes (*DCHS1*, *GLIS1*, *RNU11*, *PHLDA1*, and *HOXB-AS1*) were significantly altered in both comparisons of mCAFs/nmCAFs and mCAFs/NOFs (Figure 1C). In particular, two genes (*DCHS1* and *GLIS1*) were highly overexpressed in mCAFs compared to nmCAFs (Figure 1D).

Therefore, we performed RT-PCR and Western blot analysis for DCHS1 (FC:14.44) and GLIS1 (FC:10.25), the two most significantly upregulated genes in mCAFs relative to nmCAF to validate the microarray data. Both mRNA and protein expression levels of GLIS1 were proven to be significantly higher in mCAFs than in NOF and nmCAFs (Figure 1E,F). In addition, primary cancer cells did not express DCHS1 and GLIS1 mRNA and protein. However, the protein expression of DCHS1 was not correlated with its mRNA expression, and the protein expression of DCHS1 in mCAF compared with NOF was not significantly higher (Appendix A), suggesting that post-transcriptional modification might occur in this gene.

We therefore selected GLIS1 for further study to evaluate its functional roles (e.g., migration, invasion, and angiogenesis), which might drive metastatic progression in OSCs.

### 2.2. Knockdown of GLIS1 in CAFs Suppresses Migration, Invasion, and Angiogenesis of OC Cells

To elucidate whether CAF-derived GLIS1 plays a role in the metastasis of OC cells, we introduced siRNA for GLIS1 in mCAFs. After confirming that mRNA and protein expression of GLIS1 efficiently decreased after GLIS1 siRNA transfection (Appendix A), invasion and migration assays were performed in OC cells.

We examined the invasion ability of SKOV3 and A2780 OC cells after OC cells were put into an upper chamber and CAFs were seeded into the lower chamber. Then, OC and CAFs cocultured for 6h. Both SKOV3 and A2780 cells cocultured with GLIS1-knockdown CAFs had significantly reduced invasion ability compared with cells cocultured with siControl CAFs (Figure 2A).

Next, we examined the effects of GLIS1 derived from CAFs on cell migration by wound healing assay. We prepared conditioned medium (CM) of cultured CAFs that were transfected with siControl or siGLIS1. Both SKOV3 and A2780 cells cultured with CM of siGLIS1-CAFs for 48 h had significantly reduced wound healing ability compared with siControl CAF (SKOV3: 66.2% vs. 98.1%, *p* < 0.05, A2780: 35.5% vs. 62.8%, *p* < 0.05) (Figure 2B). The siGLIS1 CM also significantly reduced adhesion of SKOV3 cells on the culture plate (57.3% vs. 104.6%, *p* < 0.001) as compared to the siControl (Figure 2C). Cell-to-cell interactions facilitating tumor cell adhesion are essential steps in the metastatic cascade [11]. Cancer cells, especially with the highly metastatic potential, are believed to have enhanced adhesion ability that facilitates the migration of the cells to a new site to establish new tumors in the body. Therefore, the cell adhesion assay is often used to evaluate the metastatic ability of cancer cells. These data suggest that GLIS1 from CAFs regulates OC cell invasion, migration, and adhesion which are related to the initial stage of metastasis.

To determine whether GLIS1 in CAFs regulates tumor angiogenesis, we examined its effect on tube formation ability of HUVECs cultured with CAF-CM. HUVECs cultured for 4 h with CM from siGLIS1-CAFs had significantly fewer capillary-like branch points than those treated with CM from siControl-CAFs (31.6% vs. 116.6%, *p* < 0.05) (Figure 2D). These data indicate that the downregulation of GLIS1 in CAFs reduces angiogenesis.

We also examined cancer cell proliferation using the CCK-8 assay to determine whether GLIS1 in CAF is involved in cancer cell proliferation. The proliferation of SKOV3 cells was significantly inhibited siGLIS1-CM (51.3%) as compared to the siControl-CM (Figure 2E). These data suggest that GLIS1 in CAFs plays a role in cancer cell proliferation.

### 2.3. GLIS1 Silencing Reduces the Peritoneal Spread of OC Cells In Vivo

To investigate the effects of CAF-derived GLIS1 on intraperitoneal tumor metastasis in vivo, we established xenograft tumor models through inoculation of a mixture of SKOV3 cells and CAFs transfected with siGLIS1 or Control siRNA at a 1:4 ratio in the peritoneal cavity of nude mice (Figure 3A). Whereas the body weight of siGLIS1 groups was steadily increased, the body weight of the mice of siControl group increased up to 9 weeks but drastically reduced in the 10th week (siControl vs. siGLIS1.; 113% vs. 133%, *p* < 0.05) (Figure 3B). We cannot exactly explain the reason why the body weight of siControl group reduced drastically at the 9th and 10th weeks, however we guess that increasing tumor burden presumably contributed to the increase in body weight to some extent until 9th week in siControl group. After 9 weeks, the mice with high tumor burden seemed to be struggling to endure them and turned out to show the loss of weight. At the end of experiment, mice were euthanized, and the tumor nodules were immediately removed and weighed (Figure 3C). The total weight of tumor nodules in the siGLIS1-CAF group was less than that of those in the siControl or Control group (39.9% and 25.6%, respectively) (Figure 3E), although it was not statistically significant. The number of tumor nodules was significantly lower in the siGLIS1 group than in the siControl group (siControl vs. siGLIS1; 20 vs. 8.6, *p* <0.05) (Figure 3F). These results indicate that GLIS1 overexpression in CAFs might be related to the peritoneal spread of OC cells and that the downregulation of GLIS1 in CAFs might inhibit peritoneal tumor spread in OC in vivo.

## 3. Discussion

CAFs are considered important factors for enhancing tumor progression by interacting with cancer cells in the TME [1,2,9,12,13,14]. Recently, there have been many efforts to identify CAF-mediated genes that regulate the progression and metastasis of cancer cells in the TME [15,16,17,18]. The relationship of cancer cells and CAFs in metastasis has been demonstrated, but the exact mode of action by which CAFs in the TME promote metastasis is unclear. Therefore, in this study, to identify the CAF-mediated genes that are responsible for metastasis in OCs, we performed a microarray analysis and identified the DEGs between mCAFs and nmCAFs isolated from OC tissues. We found that GLIS1 was the most significantly overexpressed in mCAFs, and GLIS1 derived from mCAFs is essential for cancer cell invasion and migration by using in vitro assay and in vivo xenograft experiments.

OSC, especially high-grade OSC (HGOSC), is an extremely aggressive type of ovarian cancer and accounts for 70–80% of ovarian cancer deaths due to the high metastatic potential of HGOSC [19]. Yet, despite tremendous research efforts to unravel the determinants of metastatic spread of HGOSC, effective therapies for cancer that has metastasized to other organs are often lacking. A recent study examined the variation in mutational concordance and metastatic progression of HGOSC and identified metastatic-specific events associated with gene enrichment in genes related to the regulation of the Wnt/β-catenin pathway by performing multi-region whole-exome sequencing using HGOSC primary tumors and their metastases [20]. Although they have investigated the genetic heterogeneity and evolutionary history of HGOSC and matched distant metastases, little is known about the difference in gene expressions of CAFs from metastatic and nonmetastatic OSC.

It has been reported that CAFs enhance the metastatic ability of ovarian cancer cells with increased metastatic nodules in the peritoneal cavity. In the intraperitoneal metastatic microenvironment, CAFs govern the metastatic cascade, including the adhesion, proliferation, invasion, and colonization of metastatic sites via increasing production of several molecules (CXCL12, IL-6, VEGFA, TGF-α, and β, FGF-1) within the tumor microenvironment [21,22,23,24]. Some previous studies compared CAF and normal fibroblasts and identified CAF-associated genes, such as SRPX and HMCN1, which were experimentally related with OC migration and invasion [25,26]. However, in the present study, we compared three types of fibroblasts (mCAFs, nmCAFs, and NOFs) to find the metastasis-related genes and identified GLIS1 through the microarray analysis and validation with RT-PCR and Western blotting. GLIS1 was the most differentially upregulated gene in mCAFs compared with nmCAFs at both the mRNA and protein levels. DCHS1 was also a noticeable gene. However, in validation study with Western blotting (Appendix A), the protein expression of DCHS1 in mCAF was not significantly higher than NOF, although it was higher than nmCAF. DCHS1 might be an interesting target for metastasis because it is a member of cadherin and is involved in cell–cell adhesion [27]. Further studies would be needed to elucidate whether DCHS1 is involved in CAF-mediated cancer metastasis or not.

GLI-similar (GLIS) proteins constitute a subfamily of Krüppel-like zinc finger proteins, one of the largest families of transcription factors involved in the regulation of many cellular processes, including oncogenesis [28,29]. GLIS proteins act either as activators or repressors of gene transcription by recognizing a G-rich DNA-binding sequence, referred to as a GLIS-binding site (GLISBS), in the regulatory regions of target genes. GLIS1 and GLIS3 enhance the reprogramming of fibroblasts during induced pluripotent stem cell generation [30,31,32], suggesting their roles in cellular differentiation, proliferation, and stem cell renewal. The GLIS DNA-binding domain exhibits high homology with members of the closely related glioma-associated GLI subfamily, whose transcription factors are part of the Hedgehog signaling pathway, which is implicated in the initiation and maintenance of many cancers [33,34]. A recent study demonstrated that GLIS1-PAX8 or GLIS3-PAX8 rearrangement in hyalinizing trabecular tumor, a rare type of thyroid tumor [35]. However, the physiological roles of GLIS1 in cancer are just beginning to be recognized.

In regard to tumors, GLIS1 in various cancer cells might be involved in migration, invasion, and epithelial–mesenchymal transition (EMT), as well as oncogenesis. A previous study revealed that GLIS1 is highly expressed in several cancer cells, notably breast cancer cells, with WNT gene expression which correlated the epithelial-to-mesenchymal transition (EMT) signature. In their study, cotransfection experiments demonstrated that GLIS1 and CUX1 cooperated to stimulate TCF/β-catenin transcriptional activity and subsequently increased cell migration and invasion [36]. Another study showed that the hypoxia-inducing factors HIF2α, together with JUN, regulated GLIS1 transcription in various cancer cells [37]. GLIS1 was also reported to be related to a worse prognosis in ALL and triple negative breast cancer [38]. Deleterious mutations in GLIS1 were identified in several cases of recurrent acute lymphocytic leukemia (ALL) [39], suggesting its role in relapse in patients with high hyperdiploid ALL. A recent report suggested that low miR-1-3p expression from CAFs-derived EVs contributed to the promotion of breast cancer progression and metastasis through upregulation of GLIS1 in this subset of cancer [40]. Taken together, GLIS1 overexpression in cancer cells might be involved in cancer cell migration, invasion, and metastasis in in human leukemic or breast cancers. However, there have been no previous reports about GLIS1 expression in CAF and its role on cancer cells. In the present study, we demonstrated that GLIS1 in CAF, not in cancer cell itself, induced the cancer cell migration, invasion, and metastasis in ovarian cancers. GLIS1 cannot be an only driver gene for metastasis in ovarian cancers because metastasis is a complex multistep process involving critical interactions between cancer cells and a variety of stromal components in TME, however, this gene might be a very important gene in this process according to our results.

In summary, we, for the first time, revealed that elevated GLIS1 expression in patient-derived CAFs of OSCs with metastatic potential, and that silencing GLIS1 in mCAFs reduced motility, adhesion, angiogenesis, and metastasis of ovarian cancer cells in vitro and in vivo. Although further studies for underlying mechanism are needed to verify the exact effects of GLIS1 expression of CAFs on OSC metastasis, these results give a promising indication that CAFs derived GLIS1 may function as a therapeutic target for OSCs.

## 4. Materials and Methods

### 4.1. The Isolation of Primary Cancer Cells and CAFs from OSC Tissues

Primary cancer cells and CAFs were isolated from fresh tumor tissues from 10 patients who underwent surgery for OSC at the CHA Bundang Medical Center. Tissues were digested in PBS containing 5 µg/mL collagenase type I (Sigma Aldrich, St. Louis, MO) at 37 °C for 30 min and then washed with PBS. Carcinoma cells were then collected by centrifugation at 90× *g* for 2 min. The cells were cultured in McCoy’s 5A medium (Gibco/Invitrogen, Carlsbad, CA, USA) with 10% fetal bovine serum (FBS; Invitrogen, Carlsbad, CA, USA), EGF (100 μg/mL), 100 U/mL penicillin (Invitrogen, Carlsbad, CA, USA), and 100 μg/mL streptomycin (Invitrogen, Carlsbad, CA, USA). The supernatant containing CAFs was further centrifuged at 48× *g* for 8 min, and the pellet obtained was suspended in growth medium [11].

Normal ovarian fibroblasts (NOFs) were isolated from noncancerous ovarian tissues from 3 patients who underwent hysterectomy with oophorectomy due to uterine myoma. The cells were resuspended in DMEM/F12 medium (Invitrogen, Carlsbad, CA, USA) supplemented with 15% FBS (Gibco-BRL, Grand Island, NY, USA) and then cultured at 37 °C under humidified 5% CO_2_. The purity of the fibroblasts was determined by evaluating FAP and CK7 mRNA expression levels (Appendix A), and all experiments were carried out within 3–10 passages. Informed written consent was obtained from all patients before surgery. The study was approved by the institutional review board of CHA Bundang Medical Center, CHA University (IRB no. 2016-10-010).

### 4.2. Human Cell Lines

The human ovarian carcinoma cell lines SKOV3 and A2780 were purchased from the American Type Culture Collection (ATCC, Manassas, VA, USA) and the European Collection of Authenticated Cell Cultures (ECACC, Salisbury, UK), respectively. Cells were cultured in McCoy’s 5A medium and RPMI-1640 (Invitrogen, Carlsbad, CA, USA) supplemented with 10% FBS (Invitrogen), 100 U/mL penicillin (Invitrogen), and 100 μg/mL streptomycin (Invitrogen).

### 4.3. Microarray Analysis

The cDNA microarray was performed on 3 nmCAFs and 3 mCAFs. The cDNA was obtained using the iScript cDNA synthesis kit (Bio-Rad, Reymond, WA, USA). The synthesis of target cRNA probes and the hybridization were performed using Agilent’s Low RNA Input Linear Amplification kit (Agilent Technology, Santa Clara, CA, USA). Amplified and labeled cRNA was purified on the cRNA Cleanup Module (Agilent Technology). The fragmented cRNA was directly pipetted onto assembled Human Oligo Microarrays (60 K) (Agilent Technology). The hybridized images were scanned using a DNA microarray scanner and quantified with Feature Extraction Software (Agilent Technology). Microarray results were extracted using Agilent Feature Extraction software v11.0 (Agilent Technologies). Array probes containing Flag A were filtered. The selected Processed Signal value was transformed with a logarithm and normalized with the quantile method. The statistical significance of the expression data was determined using an independent-samples *t*-test and FC, in which the null hypothesis was that no difference exists between groups. The false discovery rate (FDR) was controlled by adjusting the *p* value using the Benjamini–Hochberg algorithm. For each DEG set, hierarchical clustering was performed using complete linkage and the Euclidean distance as a measure of similarity. Microarray data that support the findings of this study have been deposited in the Gene Expression Omnibus repository with the accession codes GSE193875

### 4.4. Reverse Transcription Polymerase Chain Reaction

Total RNA was extracted from cells using TRIzol reagent (Sigma Aldrich, St. Louis, MO, USA) according to the manufacturer’s instructions. One microgram of isolated RNA was used to synthesize cDNA using the Invitrogen Superscript III First-strand Synthesis System (Invitrogen, Carlsbad, CA, USA). Gene expression was normalized to GAPDH expression and quantified using the ImageJ Gel Analysis tool. Primers used for RT-PCR are listed in Appendix A.

### 4.5. Western Blot Analysis

Western blotting was performed as previously described [12]. Signals were detected using Enhanced Chemiluminescent Detection Reagents (Amersham Pharmacia Biotech, Little Chalfont, UK). Protein expression was normalized to β-actin expression and quantified using the ImageJ Gel Analysis tool. Primary antibodies for Western blot analysis are listed in Appendix A.

### 4.6. siRNA Transfection for GLIS1 Knockdown

Validated GLIS1 siRNAs and Control siRNA (RNAi negative control duplex) were purchased from Bioneer (Daejeon, Republic of Korea). Primary CAFs were transfected with siRNAs using Viromer Blue (Lipocalyx, Weinbergweg, Germany) according to the manufacturer’s protocol. After 48 h, transfection efficiency was confirmed by RT-PCR and Western blotting (Appendix A). The cells were used to investigate invasion and migration or to prepare conditioned medium (CM).

### 4.7. Collection of CM

CAFs were seeded and incubated for 24 h in 15% FBS DMEM/F12 followed by washing in PBS and a further incubation in serum-free media for 24 h. CM was collected by centrifugation at 3000 rpm for 10 min.

### 4.8. Cell Migration and Invasion Assays

For the invasion assay, 4 × 10^4^ SKOV3 or A2780 cells cultured in serum starvation for 24 h were loaded into the top chambers of 24-well inserts (8 µm pore size; Corning, NY, USA) coated with Matrigel (BD Biosciences, San Jose, CA, USA). CAFs were seeded into the lower chamber. The migration assay was performed in the same manner as the invasion assay except Matrigel coating in the upper chamber. After 6 h incubation, the cells that had migrated and invaded toward the lower side of the membrane were stained with hematoxylin and eosin. Then, cells were counted in three different fields under a microscope. Quantification was performed with ImageJ software version 1.53.

### 4.9. Wound Healing Assay

For wound-healing analysis, SKOV3 or A2780 cells (5 × 10^5^ cells/well) were seeded and cultured in 24 well plates for 24 h. When the cells were confluent, wounds were created by scratching the cell monolayer with a white tip. Cells were cultured in a mixture of growth medium for cancer cells (McCoy’s 5A or RPMI-1640) and CAF-CM (CAF CM for the control, siControl-treated CAF CM for siControl group, and siGLIS1-treated CAF CM, respectively) at the ratio of 1:2 in each group. The migrated area was measured after 0 and 48 h using ImageJ software. The migrated area was normalized to the initial wound width and then compared with the Control sample.

### 4.10. Cell Proliferation Assay

SKOV3 cells were cultured in a mixture of growth medium and CAF-CM (1:2) in each group for 6 days. Fresh culture medium was added every 2 to 3 days. Proliferation of cells was assessed using Cell Counting Kit-8 (CCK-8) assays (Dojindo Molecular Technologies, Inc., Kumamoto, Japan) according to manufacturer’s instructions.

### 4.11. Tube Formation Assay

Tube formation was performed in 96-well plates with pre-coated Matrigel. Human umbilical vein endothelial cells (HUVECs) were seeded (5 × 10^4^ cells/well) and cultured in CAF CM and endothelial cell growth supplement at 37 °C for 4 h. After 4 h, tube formation was quantified by counting the number of branches of the tubes by using a microscope.

### 4.12. Cell Adhesion Assay

SKOV3 cells were cultured in a mixture of CAF growth medium and CAF CM (1:2) for 18 h. After then the cells were reseeded into 96-well plates and incubated for 60 min. Nonadherent cells were removed with PBS washing. The numbers of adherent cells were measured using a CCK-8 assay. The data were calculated as percentages of Control cells.

### 4.13. In Vivo Analysis with Xenograft Model of Ovarian Cancer

All procedures involving animals were approved by the Institutional Animal Care and Use Committee (IACUC 180015) and adhered to the guidelines of the Assessment and Accreditation of Laboratory Animal Care. Xenograft tumors were established in six-week-old female BALB/C nude mice (Orient Bio, Sungnam, Korea). The animals were randomized into three treatment groups (five animals in each group): (1) PBS (Control group), (2) scrambled siRNA-transfected CAFs (siControl group), and (3) siGLIS1-transfected CAFs (siGLIS1 group). The mice in the siControl or siGLIS1 group were inoculated peritoneally with a mixture of siRNA-transfected CAFs and SKOV3 cells at a CAF: cancer cell ratio = 4(3.2 × 10^7^):1(8 × 10^6^) in 100 µL PBS. Body weight was monitored three times a week, and mice were euthanized 10 weeks after cell inoculation. Visible tumor nodules were excised and weighed. The number of tumor nodules and total tumor weight were measured in each group.

### 4.14. Statistical Analysis

Data are presented as the mean ± standard deviation. Statistical analyses were performed using unpaired Student’s *t*-tests with GraphPad Prism Software v.6 (GraphPad, La Jolla, CA, USA), and a *p*-value less than 0.05 was considered statistically significant. All results were obtained from three separate experiments. Data are represented as * *p* < 0.05, ** *p* < 0.01, *** *p* < 0.001, and **** *p* < 0.0001.

## Figures and Tables

**Figure 1 ijms-23-02218-f001:**
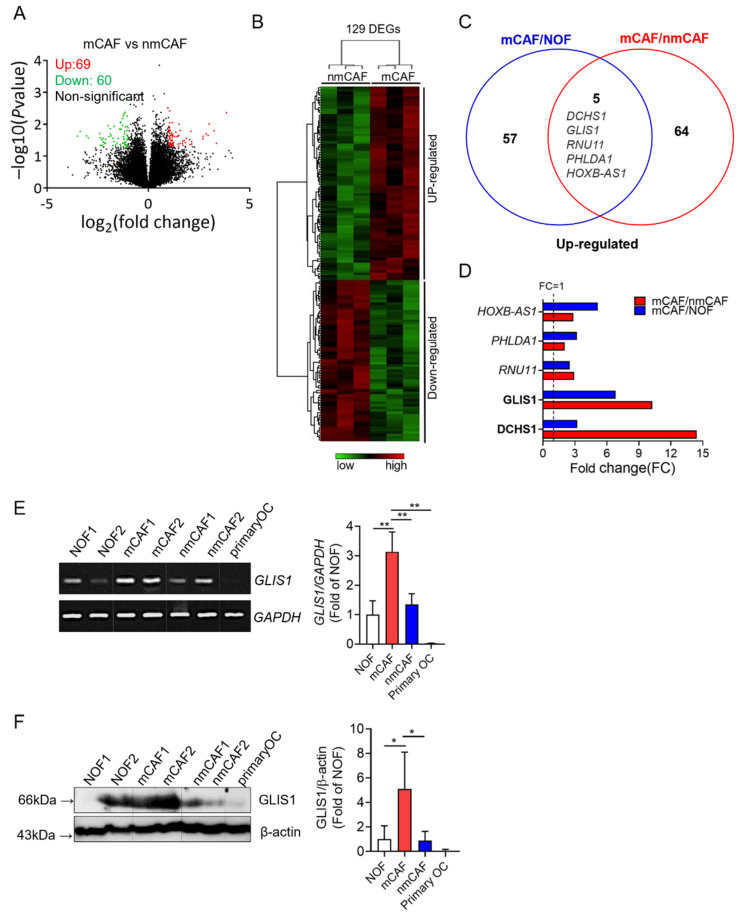
Gene expression signature of CAF from OC using microarray analysis. (**A**) Volcano plot analysis showing significantly altered genes (*p* value < 0.05, fold-change > 2 or <0.5) between mCAFs and nmCAFs; upregulated genes (red) and downregulated genes (green). (**B**) The heatmap depicts clustering of 129 differentially expressed genes on mCAFs as compared to nmCAFs (*p*-value < 0.05 and fold change > 2 or <0.5). Each column represents the expression profiles of individual tumors in each experimental group. Warm color (red) denotes an increase in gene expression, whereas cold color (green) indicates a decrease as compared to the average level of gene expression in nmCAFs. (**C**) Venn diagram showing the overlap genes significantly upregulated in mCAF vs. NOFs and mCAFs vs. nmCAFs. (**D**) Significantly upregulated five genes both mCAFs/nmCAFs and mCAF/NOF. (**E**) Expression of GLIS1 mRNA in NOF, mCAF, nmCAF, and carcinoma cells. Expression levels was examined using RT-PCR. The quantification of relative mRNA levels was normalized to GAPDH. Each experiment was performed in triplicate. Data are represented as the mean ± SD. Statistical analysis was performed using an unpaired *t*-test (** *p* < 0.01). (**F**) Immunoblot analysis of GLIS1 in NOF, mCAF, nmCAF, and carcinoma cells (primary OC). The ratio of the intensity of protein bands relative to that of β-actin was calculated. Bar graph represents the relative protein expression of GLIS1. Each experiment was performed in triplicate. Data are represented as the mean ± SD. Statistical analysis was performed using an unpaired *t*-test (* *p* < 0.05). mCAFs: metastatic CAFs, NOFs: normal fibroblasts, nmCAFs: nonmetastatic CAFs.

**Figure 2 ijms-23-02218-f002:**
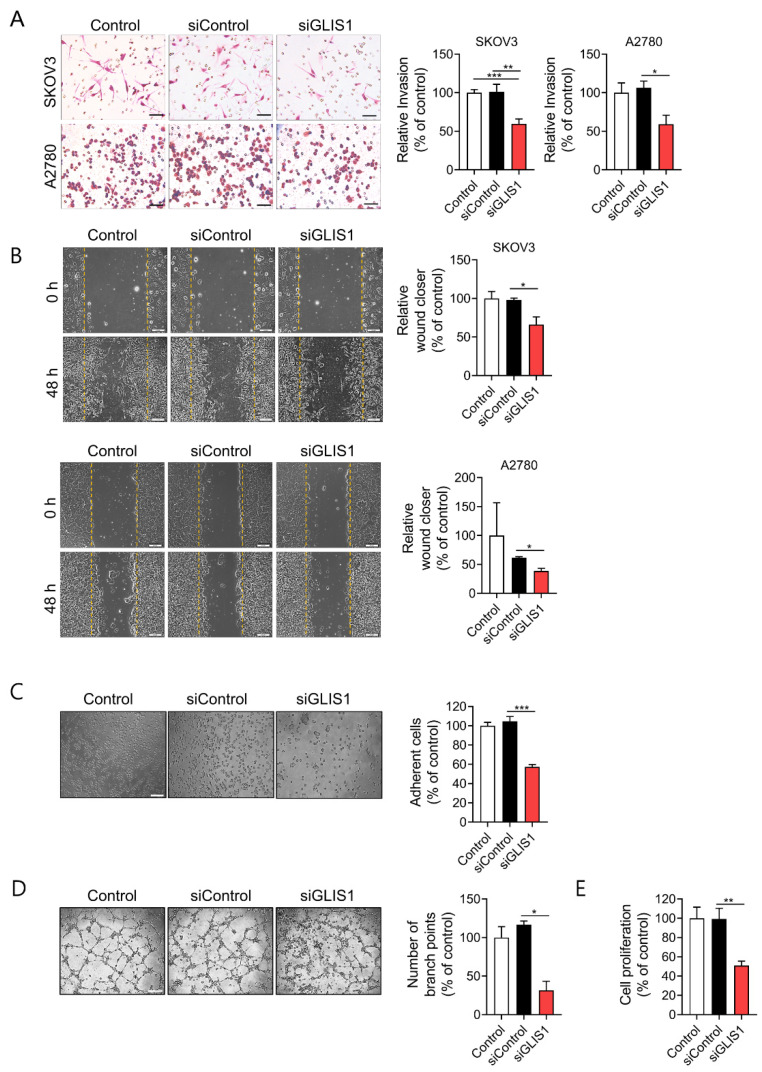
GLIS1 knockdown in CAFs inhibits OC cells migration and invasion. (**A**) Transwell Matrigel invasion assay was performed by counting migrated SKOV3 and A2780 cells co-cultured with siControl- or siGLIS1-transfected CAFs. After 6 h, the invaded cells on the basal side of the membrane were dyed and counted. Left: representative images of invaded OC cells. Right: Quantification of invasion. Percentage of number of invaded cells of each group relative to the number of the Control group. Scale bar, 50 μm. (**B**) Wound healing assays were performed to evaluate cell migration ability after 48 h. Left: representative images of scratched and recovering of wounded areas on confluence monolayers of SKOV3 and A2780 cells with CAF-CM. Yellow dotted lines indicate the wound borders at the beginning of the assay. Right: Quantification of wound closure. Relative wound closure was determined by percentage of the area of migrated cells of each group compared with that of the Control. Scale bar, 200 μm. (**C**) SKOV3 cells cultured with CAF-CM for 24 h were reseeded on plates. After 20 min, non adherent cells were removed with PBS washing. Adhesion assay was performed by counting the number of adherent SKOV3 cells. Left: representative images of adherent SKOV3 cells. Right: Quantification of adhesion. Percentage of the number of adhesive cells relative to the number of the Control group. Scale bar, 100 μm. (**D**) Tube formation abilities of human endothelial cells affected by GLIS1 silencing. HUVECs were cocultured with CAF CM and siGLIS1- or siControl-transfected CAF cells. The graph represents the relative number of branch points of HUVECs in each group compared with that of the control. Scale bar, 500 μm. (**E**) Proliferative ability, measured using a CCK-8 assay, of SKOV3 cells cultured with CAF-CM for 6 days. Percentage of OD value of each group relative to that of the Control group. Each experiment was performed in triplicate. Data are represented as the mean ± SD. Statistical analysis was performed using an unpaired *t*-test (* *p* < 0.05, ** *p* < 0.01 and *** *p* < 0.001).

**Figure 3 ijms-23-02218-f003:**
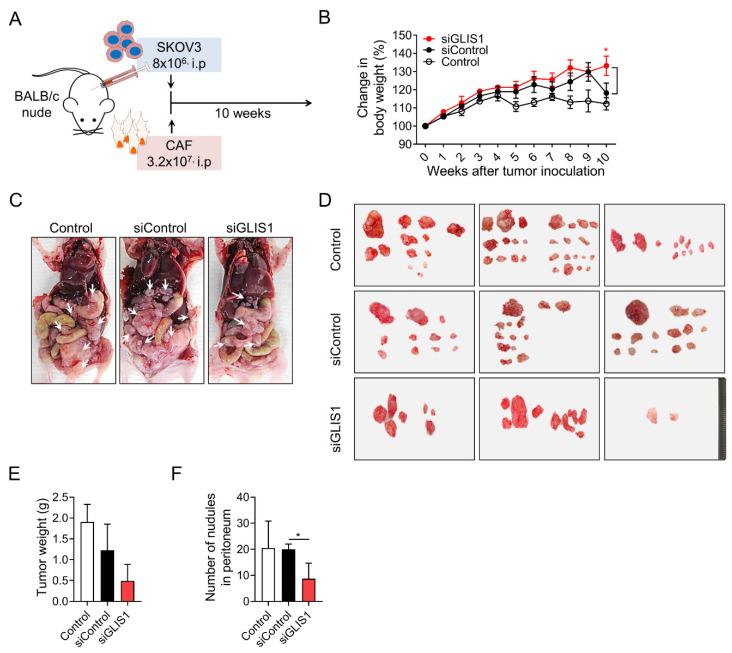
In vivo effect of GLIS1 silencing in CAFs on peritoneal tumor formation. (**A**) Schematic experimental design. SKOV3 cells mixed with CAFs transfected with siGLIS1 or Control siRNA (siControl) were intraperitoneally inoculated into nude mice. The weights and numbers of tumor nodules were observed after 10 weeks (n = 5). (**B**). The graph represents the changes in body weight from each group. (**C**) Representative image of peritoneal tumor nodules from the control, siControl, and siGLIS1 groups. Arrows indicate disseminated tumors. (**D**) Images of tumor nodules isolated from each group. Tumor weight (E) and number of nodules (**F**) were measured. Data are represented as the mean ± SD. Statistical analysis was performed using an unpaired *t*-test (* *p* < 0.05).

## Data Availability

The data presented in this study are available on request from the corresponding author.

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
