# Peer review of "GLIS1 in Cancer-Associated Fibroblasts Regulates the Migration and Invasion of Ovarian Cancer Cells"

_ijms, 2022, doi:10.3390/ijms23042218_

Round 1

Reviewer 1 Report

The paper describes the presence of specific protein (GLIS1) associated with cancer associated fibroblast (CAF) that drives migration and invasion of ovarian cancer cells. The paper is straight forward, and authors have knockdown the GLIS1 to prove their hypothesis, in in vitro and in vivo. The results are promising and support their hypothesis. I have some comments as below, before recommending the paper for publications.

  • Figure 1D shows that protein DCHS1 shows highest fold difference between mCAF/nmCAF. So, this may be also interesting protein to study and may play significant role in cancer progression. Why authors did not proceed to do knockdown study with this protein as well? Although I understand GLIS1 showed maximum difference in mCAF/nmCAF, DCHS1 might be equally interesting to see.
  • I have some concerns regarding blot full image. There is two separate image for GLIS1 and loading control b-actin. It is better if both are taken in a single image so that the exposure will stay same in both case. Since here, intensity is quantified and normalized against b-actin, it is important both images should have same exposure settings.

The full blot image for Figure 1F, DCHS1 blot background is very dark compared to white background of b-actin blot. This will manipulate the blot intensity.

Also, the blot signal are connected to eachother, which is not ideal? What is the reason here?

  • Figure 3- what is the analysis done for the confirmation of tumor nodule? A histology based confirmation of tissue is recommended. If you look at Figure 3E and 3F, between control and sicontrol, there seems significant difference in tumor weight, however number of nodules seems similar. How do authors explain this.
  • It is recommended to provide proteomics raw data in supplementary file.

Reviewer 2 Report

This article has important clinical significance. Tumor microenvironment causes therapy resistance, besides tumor cell metastasis. CAF mediated therapy resistance and role of GLIS1 needs to be discussed in the discussion section.

Other than that, I will suggest adding following information to improve the quality of the manuscript.

  1. What is the status of GLIS1 targeted therapy is other cancer? Please discuss.
  2. Please provide better quality blot image for GLIS1 and replace  Figure 1F. If GLIS1 is tumor promoting protein, please explain the probable reason of higher levels of GLIS1expression in 1 of the 2 (50%) NOFs?
  3. The growth media used for cancer cells (RPMI-1640) and the fibroblast cell (DMEM/F12) are different. Then during conditioned media experiments, how this discrepancy of the media was normalized? Please provide the data of the experimental control to address this issue.
  4. Authors have provided the method to show purity of normal fibroblasts (by FAP and CK7 expression). How they measured the purity of CAFs isolated from the tumor mass?
  5. According to Figure 1F, GLIS1 expression was not remarkable in the primary cancer cells. It was high in metastatic CAFs. Then in line 234, discussion section the authors are proposing that “ GLIS1 overexpression in cancer cell itself might be involved in cancer cell migration, invasion and metastasis.” What is the explanation of this statement?    
  6. In Figure 3D , what will be the probable cause for variation in tumor size in the control group?
  7. Line 154, please check for the typo (siControl vs siGLIS1.; 113% g 133%, p < 0.05).
  8. Figure 2F will be Figure 2E in the figure legend (line 144). Please edit that.
  9. In line 116 the authors claimed: “The siGLIS1 CM also significantly reduced adhesion of SKOV3 cells on the culture plate (57.3% vs 104.6%, p<0.001) as compared to the siControl (Figure 2C).” If the treatment reduces adhesion of the cancer cells, is it increasing anchorage independent growth? If so, it will not be an efficacious therapy at the end. Please clarify this issue.
  10. According to representative picture of Figure 2A and 2B, the results between siControl and siGLIS1 are not extremely different, although they are statistically significant. GLIS1 may be an important protein in CAFs that is promoting metastatic tumor growth, but it may not be the solo driver. That is the reason why the difference between 2 siRNA groups are not drastically different. What will be the probable other drivers for metastatic growth of ovarian cancer? Please discuss in the discussion section.    
  11. Body weight of the mice of two groups were also similar up to 9 weeks. What may be the probable reasons to drastically reduce the body weight of siControl group within 1 week (9th week to 10th week)?

Round 2

Reviewer 2 Report

Happy with the answers from the authors. In my opinion the authors may include the answers of comment # 3, 9 and 11 in the method and discussion section to further improve the manuscript.  

Author Response

Please see the attachment." 
